# Correlation of Lymphocyte Subpopulations, Clinical Features and Inflammatory Markers during Severe COVID-19 Onset

**DOI:** 10.3390/pathogens12030414

**Published:** 2023-03-06

**Authors:** Angelos Liontos, Alexandros-George Asimakopoulos, Georgios S. Markopoulos, Dimitrios Biros, Lazaros Athanasiou, Stavros Tsourlos, Leukothea Dova, Iro-Chrisavgi Rapti, Ilias Tsiakas, Evangelia Ntzani, Evangelos Evangelou, Ioanna Tzoulaki, Konstantinos Tsilidis, George Vartholomatos, Evangelia Dounousi, Haralampos Milionis, Eirini Christaki

**Affiliations:** 11st Division of Internal Medicine & Infectious Diseases Unit, University Hospital of Ioannina, Faculty of Medicine, University of Ioannina, 45110 Ioannina, Greece; 2Clinical and Molecular Epidemiology Unit, Department of Hygiene and Epidemiology, School of Medicine, University of Ioannina, 45110 Ioannina, Greece; 3Haematology Laboratory, Unit of Molecular Biology and Translational Flow Cytometry, University Hospital of Ioannina, 45110 Ioannina, Greece; 4Department of Nephrology, University Hospital of Ioannina, Faculty of Medicine, University of Ioannina, 45110 Ioannina, Greece

**Keywords:** COVID-19, T lymphocytes, lymphocyte subpopulations, inflammatory biomarkers

## Abstract

Background: Dysregulation of the immune response in the course of COVID-19 has been implicated in critical outcomes. Lymphopenia is evident in severe cases and has been associated with worse outcomes since the early phases of the pandemic. In addition, cytokine storm has been associated with excessive lung injury and concomitant respiratory failure. However, it has also been hypothesized that specific lymphocyte subpopulations (CD4 and CD8 T cells, B cells, and NK cells) may serve as prognostic markers for disease severity. The aim of this study was to investigate possible associations of lymphocyte subpopulations alterations with markers of disease severity and outcomes in patients hospitalized with COVID-19. Materials/Methods: A total of 42 adult hospitalized patients were included in this study, from June to July 2021. Flow-cytometry was used to calculate specific lymphocyte subpopulations on day 1 (admission) and on day 5 of hospitalization (CD45, CD3, CD3CD8, CD3CD4, CD3CD4CD8, CD19, CD16CD56, CD34RA, CD45RO). Markers of disease severity and outcomes included: burden of disease on CT (% of affected lung parenchyma injury), C-reactive protein and interleukin-6 levels. PO2/FiO2 ratio and differences in lymphocytes subsets between two timepoints were also calculated. Logistic and linear regressions were used for the analyses. All analyses were performed using Stata (version 13.1; Stata Corp, College Station, TX, USA). Results: Higher levels of CD16CD56 cells (Natural Killer cells) were associated with higher risk of lung injury (>50% of lung parenchyma). An increase in CD3CD4 and CD4RO cell count difference between day 5 and day 1 resulted in a decrease of CRP difference between these timepoints. On the other hand, CD45RARO difference was associated with an increase in the difference of CRP levels between the two timepoints. No other significant differences were found in the rest of the lymphocyte subpopulations. Conclusions: Despite a low patient number, this study showed that alterations in lymphocyte subpopulations are associated with COVID-19 severity markers. It was observed that an increase in lymphocytes (CD4 and transiently CD45RARO) resulted in lower CRP levels, perhaps leading to COVID-19 recovery and immune response homeostasis. However, these findings need further evaluation in larger scale trials.

## 1. Introduction

SARS-CoV-2 infection leads to activation of the immune system. Activated B lymphocytes through the production of antibodies (IgM, IgA and IgG) along with activated CD4+ and CD8+ T lymphocytes respond to viral entry and finally lead to protective immunity [1,2]. This antibody-mediated immunity pathway led to the development of effective vaccines against SARS-CoV-2 infection and the use of convalescent plasma with neutralizing capability from recovered COVID-19 patients in the initial stages of the pandemic [3,4,5]. On the other hand, specific CD4+ and CD8+ T cell subpopulations exert a key role in T cell-mediated immunity [2,6,7]. T lymphocytes exert a key role in viral infections. CD4+ T cells orchestrate the immune response and can help B cell stimulation, resulting in antibody production. CD8+ T cells, through their cytotoxic action, eliminate infected cells, which consequently leads to a reduced viral load [2].

However, dysregulation of the immune response can lead to severe forms of COVID-19 disease [8]. In these cases, an increase in serum inflammatory cytokine levels [e.g., interleukin-6 (IL-6)] is commonly observed [9,10,11,12]. Furthermore, several studies have reported the presence of lymphopenia (especially in CD4+ and CD8+ T cells) in COVID-19 patients. Of note, lymphopenia has been identified as a common characteristic in moderate and severe COVID-19. In addition, several studies have shown that decreased CD8+ T cell levels are associated with worse COVID-19 outcomes, in terms of severity and mortality [12,13,14,15,16,17]. On the other hand, patients with mild symptomatology often present with normal or elevated counts of T cell subpopulations [15,18].

Despite the growing evidence regarding the mechanisms of this maladaptive immune response, the association between cytokine levels and lymphocyte subpopulations’ regulation remains largely unclear. The aim of this study was to describe the association of lymphocyte subpopulations with markers of disease severity and outcomes in adult hospitalized COVID-19 patients.

## 2. Materials and Methods

### 2.1. Study Population

Patients were hospitalized in the Infectious Diseases Unit of the University Hospital of Ioannina. Adult patients with COVID-19 disease hospitalized between June and July 2021 were randomly selected for inclusion in the study. SARS-CoV-2 infection was diagnosed by the reverse transcriptase–polymerase chain reaction (RT-PCR) test on nasopharyngeal swab specimens. This study was part of a larger COVID-19 hospitalized patient cohort study, which was approved by the Institutional Ethics Committee of the University Hospital of Ioannina [Protocol Number: 5/11-03-2021 (issue:3)/The University Hospital of Ioannina COVID-19 Registry, NCT05534074]. Epidemiological, clinical, and laboratory parameters were obtained from the University Hospital of Ioannina COVID-19 Registry. Data were imported in a digital database anonymously with a personal identifier code for each patient, as prespecified by study protocol. Data collection was conducted following the highest standards of European Guidelines for Good Clinical and Laboratory Practice in Research Studies/Protocols and in accordance with the Helsinki Declaration. Each patient included in this study provided a written informed consent.

Patient demographics and duration of symptoms (days) were documented on admission (day 1). Burden of lung disease on Computed Tomography (CT) was defined as the percentage (%) of affected lung parenchyma. Calculation of lung involvement was made similarly to Chung et al. [19]. In this study, each of the five lung lobes was assessed for the degree of involvement and classified as none (0%), minimal (1–25%), mild (26–50%), moderate (51–75%), or severe (76–100%) [19]. In the present study lung involvement was calculated based on 2 axes: (1) in the evaluation of the entire lung parenchyma, simple approximation was as follows: 25% of lung parenchyma referred to each one of the lower lobes while 15% referred to each upper and middle lobe, respectively; (2) classification of severity was defined as: minimum (<10%), moderate (11–25%), important (26–50%), severe (51–75%), and critical (>75%), respectively.

C-reactive Protein (CRP), interleukin-6 (IL-6) levels, partial O2 pressure/fraction of inspired O2 (PO2/FiO2 ratio), and lymphocyte subpopulations were measured at two timepoints (day 1 and day 5). IL-6 was calculated by Access IL-6 assay, a paramagnetic particle, chemiluminescent immunoassay executed on Beckman DXi 800 immunoassay analyzers. CRP was calculated by rate nephelometry (IMMAGE, Beckman Coulter). Also, differences in lymphocyte subpopulations and inflammatory markers between the two timepoints were calculated. Patients’ demographic, laboratory and clinical characteristics were documented by timepoint as mean [± standard deviation, (SD)] or frequency rates.

Since the early days of the pandemic, it has been shown that worsening of COVID-19 occurs from day 7 through day 12 of the disease. Most of the patients who deteriorated, did so by day 7 [20]. In the study cohort, the mean duration of symptoms (from disease onset) was 6.29 days. Therefore, the rationale for measuring lymphocyte subpopulations on day 5 of hospitalization, as a second timepoint, was to include the timeline of clinical deterioration and not exceed the average length of stay, to reduce selection bias.

### 2.2. Flow Cytometry Analysis

Specific lymphocyte subpopulations were calculated with the use of flow-cytometry on admission (day 1) and at day 5 of hospitalization, based on established protocols [21]. Briefly, analyses were performed in a FACScalibur cytometer, using CellQuest V3.1 software (both by Becton Dickinson). Lymphocyte subpopulations (CD45) from peripheral blood include B (CD19) and T (CD3) lymphocytes, CD4 T helper cells, CD8 T cytotoxic cells, NK (natural killer) cells CD16+56 cells, as well as CD45RA+ (naïve) CD45RA+RO+ (transient) and CD45RO+ (memory) cells.

### 2.3. Statistical Analysis

Logistic and linear regression adjusted for age, sex, and duration of symptoms were used in the analyses of categorical and continuous outcomes for each subpopulation of cells, respectively. False density rate (FDR, Benjamini–Hochberg) was used to correct for multiple testing error. Violin plots were introduced to visualize the distribution of the variables of the most significant associations. All statistical analyses were performed using the STATA 17 (Stata Corp LP, College Station, TX, USA) software.

## 3. Results

A total of 42 hospitalized patients were included in the study. Patients’ descriptive characteristics on admission are shown in Table 1, as an entire cohort and stratified by lung injury status. Female patients represented the majority of the study population (69.01%, *n* = 29). The mean age was 55.9 (SD: 20.3) years. Arterial hypertension and obesity were the most common comorbidities documented in the study (*n* = 21 and 14, respectively). Of note, in the studied population none of the patients had comorbidities such as rheumatoid arthritis, systemic lupus erythematosus, inflammatory bowel disease, or other chronic immune-mediated/autoimmune diseases that may bias the interpretation of the lymphocyte subpopulations. Of note, none of these patients had a history of immuno-modulatory medication use, either.

Lung injury ≤50% was observed in 19 patients (55.48%), while 15 patients (44.12%) had greater injury (>50%) of lung parenchyma. Mean (SD) IL-6 was 39.46 IU/mL (52.48) at admission. Mean (SD) C-reactive protein was 72.90 mg/L (70.98) and 27.93 mg/L (40.09) on day 1 and day 5, respectively. Mean (SD) values of PO2/FiO2 were 249.59 mmHg (142.52) at baseline and 209.29 mmHg (135.93) at day 5. Similar reductions were observed for the aforementioned variables between day 1 and 5 for both ≤50 and >50% of lung injury. Results of linear and logistic regression analyses are shown in Table 2 and Table 3.

The most significant associations were between lymphocyte cell subpopulations and the extent of lung injury (CT burden of disease) and CRP levels. CD16+56 cells (NK-cells) were associated with higher risk of lung injury (OR = 1.193; 95% CI: 1.019 to 1.397; *p* = 0.028). On the other hand, an inverse association was observed for CD3+CD8+ cells (OR = 0.855; 95% CI: 0.741 to 0.986; *p* = 0.032). However, after FDR correction, none of these associations remained statistically significant. All other subpopulations did not show a significant association with these markers of disease severity.

Further analysis showed that an increase per 1% in CD3+CD4+ and CD4RO+ cell count difference between day 5 and day 1 resulted in a negative CRP difference between these time points (lower CRP value at day 5 than day 1, Beta = −5.227; 95% CI: −10.353 to −0.099; *p* = 0.046 and Beta = −5.327; 95% CI: −9.715 to −0.938; *p* = 0.020, respectively). On the other hand, CD45RA+RO+ cell count difference was associated with a positive difference between CRP levels at the two timepoints (greater CRP value at day 5 than day 1, Beta = 4.661; 95% CI: 0.631 to 8.689; *p* = 0.026). However, similarly, this association was no longer significant after FDR correction. No significant association was found in the kinetics of other subpopulations of lymphocytes and the inflammatory markers. Similarly, no significant association was found in the statistical analysis between comorbidities and the degree of lung injury (all *p* = NS, data not shown).

Figure 1 and Figure 2 show the distribution of CRP with significant cell markers (before FDR correction) at day 1 and day 5 (Figure 1), as well as the distribution of CRP and lymphocyte count differences between these timepoints (Figure 2).

## 4. Discussion

This study evaluated the association of different lymphocyte subpopulations with COVID-19 disease severity markers in hospitalized patients. Several previous studies in COVID-19 patients have assessed the possible role of lymphocyte cell populations as predictors and markers of disease severity and outcome.

Common viral infections typically lead to expansion of T cells and a resulting lymphocytosis. T cell cytotoxicity exerts a key role in the elimination of infected cells in COVID-19, similar to other viral infections. However, in these patients, the lymphocyte count and especially CD8+ T cells are greatly reduced [15,22]. It has been shown that this reduction is strongly associated with the severity of COVID-19 [15,23,24]. In the study by Ashrafi et al., (*n* = 40) the decreased number of T cells and particularly the CD4+ T cell count were associated with higher mortality rates due to severe COVID-19 [25]. This finding can be interpreted by the central role of CD4+ helper T cells in immunity [26]. In another study, CD8+ T lymphopenia among all T cell subpopulations was a more profound factor in disease severity progression [27]. In addition, in the meta-analysis by Huang et al. it was observed that absolute counts of lymphocyte subpopulations were decreased in severe cases of COVID-19, compared to less severe disease [28]. CD16+56 NK cell counts have been shown to decrease in COVID-19 disease [29]. Cytolytic activity of NK cells and associated production of cytokines exert a key role in the defensive immunological response against SARS-CoV-2 infection [30]. This cytolytic activity of NK cells serves as a possible mechanism of progressive lung damage observed in the present study with the increase in NK count. Similarly, in a study (*n* = 32 patients) with severe SARS-CoV-2 infection, a raised proportion of mature NK cells and low T cell counts was observed [31].

In a study of 103 COVID-19 patients from the early stages of the pandemic, it was shown that CD3+ T cells, CD4+ T cells, CD8+ T cells, and CD16+56 NK cell counts were significantly decreased in these patients compared to healthy individuals (controls) and were inversely associated with the severity of the infection. Of note, it was also observed that low CD8+ T levels were more common than low CD4+ T levels in COVID-19 patients [32]. Similarly, in a single-center, retrospective study (*n* = 164), it was shown that decreased counts of total CD3+, CD3+CD8+, CD3+CD4+, and CD3+CD4+CD8+ T lymphocytes were associated with higher rates of in-hospital mortality [33]. Aside from their role as predictors of disease severity, the measurement of T cell subpopulations has also been examined in distinguishing COVID-19 from other lung infections, such as community acquired pneumonia (CAP), even during the early phase of the pandemic. In a single-center study of 296 confirmed COVID-19 cases and 130 CAP cases, analysis showed that mean values (% counts) of CD19+ and CD3+CD8+ in COVID-19 patients were significantly lower than in patients with CAP. On the other hand, mean values (absolute and % counts) of CD3+CD4+, CD16+56+ in COVID-19 patients were significantly higher than in patients with CAP [34]. Furthermore, several studies have proposed a protective role of CD8+ T cells in the progression to more severe forms of COVID-19, especially in the acute phase of the disease [35,36].

In our study, a higher CD3+ T cytotoxic (CD3+CD8+) cell count on admission was associated with a lower risk of lung injury >50% as assessed by CT, resulting in a less severe form of the disease. Of note, our study population represents a rather moderate-severity cohort of patients, as evidenced by a mean PO2/FiO2 ratio of 294.59 on admission/day 1. Another point of interest is that the mean duration of symptoms in our patients was six days. According to a previous study, the decrease of T cells in severe cases of disease was noted at its highest point within the first seven days from symptoms’ onset. Eventually, T cells were gradually restored to normal levels after the third week from the onset of symptoms [15].

IL-6, among cytokines, has been shown to be associated with more severe forms of COVID-19 [23], probably as a result of the induced cytokine storm [37]. In a large meta-analysis of 7865 patients, a decrease in lymphocytes and elevated IL-6 levels were found in the group of severe cases compared to mild disease [38]. However, in the present study, no statistical association was found between IL-6 and lymphocytes. Similarly, no statistical association was observed in linear regression analysis of the change between day 5 and day 1 of PO2/FiO2 with the change in cell subpopulations. The lack of significance for IL-6 and PO2/FiO2 in our cohort may be explained by the less severe form of disease from admission to day 5, and the rather limited sample-size.

It is evident that CRP is a strong indicator of the inflammatory process and a predictor of COVID-19 disease severity [23,24,39,40,41]. However, data regarding the association of lymphocytes and CRP levels are scarce. In a study of 172 COVID-19 hospitalized patients, a significant correlation between CD8+ T cells and inflammation markers such as CRP and Neutrophil to Lymphocyte ratio (NLR) was observed [42]. Also, a small-sized study (*n* = 33 patients), showed a negative correlation between CRP levels and T cell subpopulations. Specifically, CRP was inversely associated with CD3+, CD3+CD4+ and CD3+CD8+ T cells (rho = −0.77, *p* < 0.001, rho = −0.74, *p* < 0.001, rho = −0.66, *p* = 0.001), respectively [43].

Changes in lymphocyte subpopulations have been observed in patients with COVID-19, and these changes have been linked to the severity of the illness. During the onset of COVID-19 disease, a polarization of lymphocytes towards a memory phenotype occurs, in which naïve CD45RA cells are flowing in a transient state (CD45RA+RO+) to become memory (CD45RO) cells [44]. This polarization is accompanied by a functional exhaustion of immune cells [45], and leads to the reduction of both cellular [46] and humoral immune responses [47]. Importantly, the transition between naïve (CD45RA) to memory (CD45RO) cells is a prerequisite for a successful immune response that is associated with a milder COVID-19 onset and course [48]. The associations found in this study confirm previous data and support that during hospitalization, a clinical improvement is associated with this transition, which is inversely proportional to CRP levels. Specifically, the reduction of CRP between day 1 to day 5 is associated with an induction of CD45RO subpopulation, indicating an initiation of immune response. The impact of lymphocyte populations in COVID-19 is complex and dependent on the stage of the illness and the individual patient’s immune response. Further research is needed to fully understand the role of lymphocytes in COVID-19 and how to best use this information to improve treatment and outcomes for patients.

Our study has some limitations. First, the sample size is small, and the study involves only one medical center, which, however, covers a wide geographical area of western Greece. Second, comparisons between subgroups of patients based on the degree of lung injury did not yield statistically significant results, owing to the small number of patients in each group. Third, for the same reasons, it was not possible to analyze outcomes such as death and intubation in this cohort of patients due to the small number of these events. Last, measurement of cell counts and inflammatory markers at more timepoints may have yielded more granular results in terms of lymphocyte subpopulation kinetics and their association with inflammatory markers.

## 5. Conclusions

Despite its limited size, our study showed that alterations in lymphocyte specific subpopulations are associated with COVID-19 severity. Also, the observed increases in memory T cells (CD45RO) and CD4 T cells (CD3+ and RO+) were associated with lower CRP levels, potentially leading to COVID-19 recovery and immunity. However, these findings need further evaluation and validation in larger scale trials, in order to draw conclusions on the role of specific lymphocyte subpopulations in COVID-19 progression.

## Figures and Tables

**Figure 1 pathogens-12-00414-f001:**
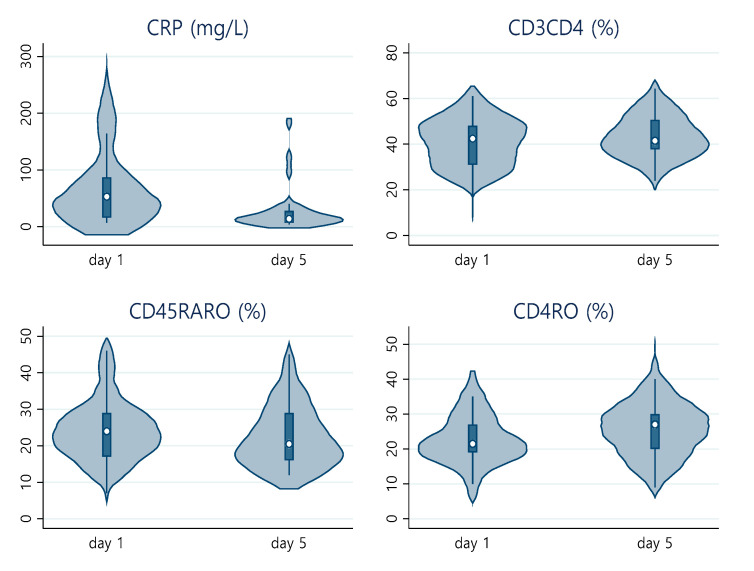
Violin plot of CRP CD3+CD4+, CD45RA+RO+ and CD4RO in day 1 and 5.

**Figure 2 pathogens-12-00414-f002:**
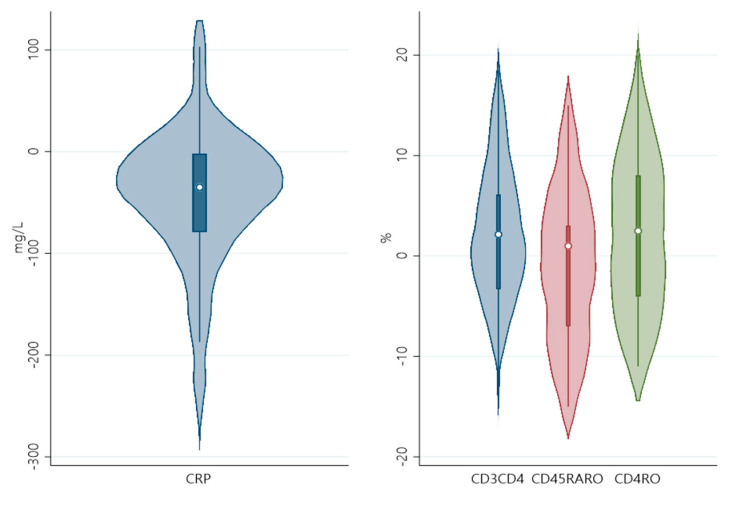
Violin plot of differences between timepoints of CRP and CD3+CD4+, CD45RA+RO+, CD4RO.

**Table 1 pathogens-12-00414-t001:** Descriptive characteristics of variables.

Characteristics	N	Mean (SD) or Frequency	CT Burden of Disease	
			≤50% (*n* = 19)	>50% (*n* = 15)
Age, years, mean (SD)	42	55.90 (20.30)	54.63 (16.69)	60.13 (19.16)
Sex, *n* (%)	42			
Female		29 (69.05)	15 (78.95)	9 (60.00)
Male		13 (30.95)	4 (21.05)	6 (40.00)
Duration of symptoms, days, mean (SD)	38	6.29 (3.69)	6.83 (4.00)	6.35 (3.69)
Comorbidities, *n* (%)	42			
Arterial Hypertension		21 (50.00)	12 (63.15)	8 (53.30)
Diabetes Mellitus		7 (16.70)	3 (15.80)	3 (20.00)
Coronary Artery Disease		6 (14.30)	2 (10.50)	2 (13.30)
Stroke		1 (2.40)	0	1 (6.60)
Cancer (non-active)		2 (4.80)	0	2 (13.30)
Obesity		14 (33.30)	9 (47.40)	4 (26.60)
Chronic Obstructive Pulmonary Disease		1 (2.40)	1 (5.30)	0
Smoking (Active)		2 (4.80)	2 (10.50)	0
Dyslipidemia		7 (16.70)	3 (15.80)	2 (13.30)
Day 1				
IL-6, IU/mL, mean (SD)	32	39.46 (52.48)	27.43 (21.27)	60.73 (79.73)
C-reactive protein, mg/L, mean (SD)	42	72.90 (70.98)	70.05 (67.10)	85.73 (85.06)
PO2/FiO2, mean (SD)	41	294.59 (142.52)	342.67 (138.76)	244.15 (117.46)
CD45, mean absolute number	42	1259 (811.88)	1248.21 (712.66)	923 (538.01)
CD3 %, mean (SD)	42	66.69 (11.69)	65.04 (10.82)	64.86 (10.56)
CD3CD8 %, mean (SD)	42	25.04 (10.38)	26.46 (10.83)	19.28 (6.22)
CD3CD4 %, mean (SD)	42	40.39 (10.85)	37.65 (10.84)	45.03 (10.46)
CD3CD4CD8 %, mean (SD)	42	1.17 (1.12)	1.25 (1.45)	1.14 (0.89)
CD16+56 %, mean (SD)	42	19.33 (9.20)	17.87 (8.86)	21.54 (8.03)
CD19 %, mean (SD)	42	13.09 (7.72)	15.97 (7.33)	12.82 (8.18)
CD45RA %, mean (SD)	42	45.02 (12.17)	46.42 (10.75)	42.20 (10.37)
CD45RO %, mean (SD)	42	30.29 (10.97)	31.52 (11.56)	31.40 (10.84)
CD45RA+RO+ %, mean (SD)	42	24.40 (8.44)	22 (6.70)	25.73 (7.98)
CD4RA %, mean (SD)	42	16.98 (11.14)	13.78 (9.46)	20.26 (12.45)
CD4RO %, mean (SD)	42	22.74 (7.62)	22.73 (8.59)	24.86 (6.85)
Day 5				
C-reactive protein, mg/L, mean (SD)	29	27.93 (40.09)	44.16 (58.83)	15.84 (10.48)
PO2/FiO2, mean (SD)	19	209.29 (135.93)	260.98 (142.70)	203.27 (138.08)
CD45, mean absolute number	34	1745 (918.89)	2042.68 (890.23)	1527.78 (778.17)
CD3 %, mean (SD)	34	67.07 (11.58)	66.10 (10.14)	68.61 (12.26)
CD3CD8 %, mean (SD)	34	22.76 (7.49)	23.66 (5.96)	20.28 (7.83)
CD3CD4 %, mean (SD)	34	43.28 (9.53)	42.16 (8.56)	47.07 (9.38)
CD3CD4CD8 %, mean (SD)	34	1.36 (1.48)	1.56 (1.93)	1.24 (1.06)
CD16+56 %, mean (SD)	34	12.65 (7.92)	12.20 (7.35)	11.48 (6.09)
CD19 %, mean (SD)	34	18.88 (10.66)	20.46 (8.81)	18.17 (12.01)
CD45RA %, mean (SD)	34	43.38 (14.55)	43.87 (13.98)	42.71 (13.32)
CD45RO %, mean (SD)	34	33.35 (11.70)	3287 (11.38)	34 (10.66)
CD45RA+RO+ %, mean (SD)	34	23.26 (8.93)	23.25 (8.51)	23.28 (8.37)
CD4RA %, mean (SD)	34	16.26 (8.08)	14.87 (7.35)	18.07 (9.61)
CD4RO %, mean (SD)	34	26.03 (8.52)	25.18 (6.15)	29.07 (9.12)

**Table 2 pathogens-12-00414-t002:** Linear and logistic regression analysis adjusted for age, sex, & duration of symptoms. Lymphocyte subpopulations, CT burden of disease and IL-6 are measured on admission (day 1). False density rate was used to correct for multiple testing error. IL-6: interleukin-6, CT: computed tomography.

	CT Burden of Disease				IL-6		
Cells	OR (95% CIs)	*p*-Value	FDR	Obs	Beta (95% CIs)	*p*-Value	Obs
CD3	0.957 (0.877, 1.044)	0.327	0.546	32	0.208 (−1.584, 2.001)	0.813	31
CD3CD8	0.855 (0.741, 0.986)	0.032	0.176	32	0.109 (−1.546, 1.764)	0.893	31
CD3CD4	1.059 (0.971, 1.155)	0.191	0.499	32	0.095 (−1.647, 1.837)	0.911	31
CD3CD4CD8	0.902 (0.474, 1.717)	0.755	0.755	32	−3.263(−17.455, 10.929)	0.640	31
CD16+56	1.193 (1.019, 1.397)	0.028	0.176	32	−0.528 (−2.638, 1.581)	0.611	31
CD19	0.951 (0.857, 1.055)	0.348	0.546	32	0.380 (−2.037, 2.799)	0.749	31
CD45RA	0.974 (0.901, 1.053)	0.518	0.633	32	−0.725 (−2.541, 1.091)	0.419	31
CD45RO	0.988 (0.920, 1.061)	0.744	0.755	32	0.337 (−1.587, 2.262)	0.721	31
CD45RA+RO+	1.072 (0.961, 1.195)	0.211	0.499	32	0.547 (−1.576, 2.670)	0.601	31
CD4RA	1.057 (0.965, 1.157)	0.227	0.499	32	−0.561 (−2.278, 1.155)	0.507	31
CD4RO	1.022 (0.925, 1.129)	0.430	0.591	32	0.960 (−1.517, 3.439)	0.433	31

**Table 3 pathogens-12-00414-t003:** Linear regression analysis: association of difference between day 5 and day 1 of CRP and PO2/FiO2 with the respective difference of cell subpopulations, adjusted for age, sex & duration of symptoms. False density rate was used to correct for multiple testing error. CRP: C-reactive protein, PO2/FiO2: partial pressure of O2/Fraction of inspired O2.

Day5-Day1	Delta CRP				Delta PO2/FiO2		
DeltaCells	Beta (95% CIs)	*p*-Value	FDR	Obs	Beta (95% CIs)	*p*-Value	Obs
CD3	−3.357(−7.720, 1.007)	0.124	0.273	25	−5.199(−13.540, 3.141)	0.197	16
CD3CD8	2.738(−5.057, 10.534)	0.472	0.577	25	−3.799(−16.438, 8.839)	0.522	16
CD3CD4	−5.227(−10.353,−0.099)	0.046	0.169	25	−0.734(−11.779, 10.311)	0.886	16
CD3CD4CD8	13.895(−8.698, 36.488)	0.214	0.392	25	17.552(−18.688, 53.793)	0.309	16
CD16+56	3.117(−0.747, 6.981)	0.108	0.273	25	5.115(−1.558, 11.789)	0.120	16
CD19	−0.898(−6.699, 4.902)	0.750	0.825	25	−2.639(−12.847, 7.569)	0.581	16
CD45RA	−1.241(−4.525, 2.044)	0.440	0.577	25	0.210(−7.014, 7.436)	0.950	16
CD45RO	−2.001(−6.110, 2.108)	0.322	0.506	25	−3.652(−11.566, 4.262)	0.332	16
CD45RA+RO+	4.661(0.631, 8.689)	0.026	0.143	25	3.827(−4.612, 12.267)	0.340	16
CD4RA	−0.464(−5.611, 4.683)	0.853	0.853	25	6.252(−2.542, 15.048)	0.146	16
CD4RO	−5.327(−9.715, −0.938)	0.020	0.143	25	−1.157(−10.605, 8.291)	0.793	16

## Data Availability

Data are available upon reasonable request.

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
