# Peer review of "Correlation of Lymphocyte Subpopulations, Clinical Features and Inflammatory Markers during Severe COVID-19 Onset"

_pathogens, 2023, doi:10.3390/pathogens12030414_

Round 1
Reviewer 1 Report
Summary:
The manuscript described the association between lymphocyte subpopulations and COVID-19 infection severity clinical measurements. The authors measured the lymphocyte subpopulations on the day of patient hospital admission and day 5 of hospitalization. These measurements were compared with markers of infection severity, such as the burden of disease on CT, C-reactive protein level, Il-6 cytokine levels, and PO2/FiO2 ratio.
Major comments:
- When comparing the increase in lymphocyte subpopulation counts with the severity of infections, It is unclear if the manuscript comparison is between lymphocyte subpopulation before hospitalization and on the day of admission.
- In the introduction section, it was mentioned that the study aimed to compare lymphocyte subpopulations and the outcome of the hospitalization. However, the hospitalization outcome was not mentioned in the manuscript.
- The violin plots in Figures 1, 2, and 3 need to be better represented. Including the CRP data set and lymphocyte counts in the same graph skewed the representation of counts, and it is hard to identify the fundamental differences between days 1 and 5.
Minor comments:
- It is acknowledged that the data set is small to conclude any generalizations. The authors should have commented on the future manuscripts on the much larger data set for the registered clinical trials mentioned in the manuscript.
Author Response
Reviewer 1
The manuscript described the association between lymphocyte subpopulations and COVID-19 infection severity clinical measurements. The authors measured the lymphocyte subpopulations on the day of patient hospital admission and day 5 of hospitalization. These measurements were compared with markers of infection severity, such as the burden of disease on CT, C-reactive protein level, Il-6 cytokine levels, and PO2/FiO2 ratio.
We would really like to thank the reviewer for the consistent and meaningful comments.
Major comments:
Comment #1
When comparing the increase in lymphocyte subpopulation counts with the severity of infections, It is unclear if the manuscript comparison is between lymphocyte subpopulation before hospitalization and on the day of admission.
Answer to Comment #1
We would like to thank the reviewer for his/her comment. In our study, comparison was made between lymphocyte subpopulations, measured on the day of admission (day 1) and on day 5 of hospitalization. There were no measurements of lymphocyte subpopulations before hospitalization. This is described in page 3 of the manuscript, under section 2.1, “Study Population…lymphocyte subpopulations were measured in two timepoints (day 1 and day 5). Also, differences in lymphocyte subpopulations and inflammatory markers between the two timepoints were calculated”. Also, it is mentioned in same page, under section 2.2, “Flow cytometry…Specific lymphocyte subpopulations were calculated with the use of flow-cytometry on admission (day 1) and at day 5 of hospitalization, based on established protocols”.
Comment #2
In the introduction section, it was mentioned that the study aimed to compare lymphocyte subpopulations and the outcome of the hospitalization. However, the hospitalization outcome was not mentioned in the manuscript.
Answer to Comment #2
We would really like to thank the reviewer for this accurate comment. In page 2 of the manuscript in the introduction section we defined the aim of the study: ‘’The aim of this study was to describe the association of lymphocyte subpopulations with markers of disease severity and outcomes in adult hospitalized COVID-19 patients’’. Also, in page 2 and 3 of Materials and Methods in section 2.1, "Study Population”, we defined “Burden of lung disease on Computed Tomography (CT) was defined as the percentage (%) of affected lung parenchyma. C-reactive Protein (CRP), interleukin-6 (IL-6) levels, partial O2 pressure/fraction of inspired O2 (PO2/FiO2 ratio)’’. Since the early stages of the pandemic, all these parameters have been used as markers of COVID-19 disease severity. Of note, in our study the limited size of study population was associated unavoidably with a limited number of “hard” outcomes such as death or intubation, precluding any statistical analysis. In this perspective we used the COVID-19 associated severity of disease parameters as outcomes in our analysis.
Comment #3
The violin plots in Figures 1, 2, and 3 need to be better represented. Including the CRP data set and lymphocyte counts in the same graph skewed the representation of counts, and it is hard to identify the fundamental differences between days 1 and 5.
Answer to Comment #3
We would really like to thank the reviewer for helping us improve the violin plots. We reformed the violin plots according to reviewer’s comment. We assume that with the new figures it is clearer to understand the fundamental differences between days 1 and 5. Former figures 1, 2 and 3 are now presented as Figure 1 in page 6, similarly former figure 4 is now presented as Figure 2, in page 7.
Minor comments:
Comment #1
It is acknowledged that the data set is small to conclude any generalizations. The authors should have commented on the future manuscripts on the much larger data set for the registered clinical trials mentioned in the manuscript.
Answer to Comment #1
We would like to thank the reviewer for this remark. We aimed to investigate the association between lymphocyte subpopulations and markers of disease severity in COVID-19 patients. Our initial thoughts were to include a larger cohort of patients from the University Hospital of Ioannina COVID-19 Registry. However due to limited resources for broader patient inclusion and lack of funding to cover the cost of flow cytometry assays, we couldn’t include a greater number of patients. We hope to be able to study a larger cohort, in the near future, in order to yield results that would have wider applicability.
Reviewer 2 Report
The paper by Liontos et al assessed the relationship between COVID-19 outcome and lymphocyte subpopulations in a small cohort of patients. There are several issues that need to be clarified:
1) Abstract, Results: the Authors should specify whether an increase or a decrease in “CD16CD56 cells (Natural Killer cells) was associated with a higher risk of lung injury (>50% of lung parenchyma)”. When looking at table 1, the overall mean of NK cell frequency is normal, albeit higher on day 1 than day 5.
2) In general, the paragraph Results should be rewritten as the findings are not reported very clearly.
3) Materials and Methods: it is crucial for correct interpretation of data to disclose whether COVID-19 patients enrolled in the study had such comorbidities as rheumatoid arthritis, systemic lupus erythematosus, inflammatory bowel disease or other chronic immune-mediated/autoimmune diseases; these conditions may bias the correct interpretations of the lymphocyte subpopulations.
4) Maybe table 1 would be more interesting if data are split according to the extent of lung involvement (i.e., £50% vs >50%), adding a column with p values between the two patient groups.
5) Likewise, it would be desirable to add in Table 1 the most common patient comorbidities and to check whether there are statistical differences between the two groups (£50 vs >50%). Previous studies have consistently shown that comorbidities have a negative impact on the outcome. Besides, for small series like this one, patient groups should be as much homogeneous as possible for correct interpretation of findings.
6) Table 1, day 5. IL-6 has been measured in only two patients, this piece of information should be omitted because it does not have any significance.
7) Materials and Methods, page 3. How lung disease involvement on computed tomography (CT) was estimated should be detailed, e.g., was it calculated according to Chung et al (Radiology 2020;295:202–207) or else?
8) Materials and Methods, page 3. What was the rationale for measuring lymphocyte subpopulations on day 5? The outcome of COVID-19 is highly variable, why did the Authors chose this time point for follow-up of lymphocyte subpopulations? The Authors should explain their motivation in Material and Methods.
9) Do the Authors have data on lymphocyte subpopulations at patient discharge from the hospital?
10) Discussion: Although the Authors found an association between NK cells and lung involvement, the significance of this finding is not discussed.
11) Finally, the Authors report that none of the associations remained significant after FDR correction. How does this impact on the overall significance of the paper?
12) English revision is necessary.
Author Response
Reviewer 2
The paper by Liontos et al assessed the relationship between COVID-19 outcome and lymphocyte subpopulations in a small cohort of patients. There are several issues that need to be clarified.
We thank the reviewer for the useful and detailed comments and suggestions.
Comment #1
Abstract, Results: the Authors should specify whether an increase or a decrease in “CD16CD56 cells (Natural Killer cells) was associated with a higher risk of lung injury (>50% of lung parenchyma)”. When looking at table 1, the overall mean of NK cell frequency is normal, albeit higher on day 1 than day 5.
Answer to Comment #1
We would like to thank the reviewer for pointing this out. Data analysis of our cohort showed that higher levels of CD16CD56 cells (Natural Killer cells) were associated with higher risk of lung injury (>50%). We have reviewed the abstract and updated the text accordingly. In page 1, under Results we have the sentence “Higher levels of CD16CD56 cells (Natural Killer cells) were associated with higher risk of lung injury (>50% of lung parenchyma)” to increase clarity. Regarding the comment for Table 1, we would like to clarify that Table 1 represents descriptive characteristics of cell subpopulations in the 2 time points. However, the statistical analysis was performed on lymphocyte subpopulations on admission at the hospital (day 1), as shown in Table 2. To make it more clear, in page 5 of the manuscript, we have edited the legend of Table 2 and it now reads “Table 2. Linear and logistic regression analysis adjusted for age, sex & duration of symptoms. Lymphocyte subpopulations, CT burden of disease and IL-6 are measured on admission (day 1).”
Comment #2
In general, the paragraph Results should be rewritten as the findings are not reported very clearly.
Answer to Comment #2
We would like to thank the reviewer for the chance to improve our Results section, with his/her comment. We have updated and extended the Results section to provide a more clear and comprehensive view of the findings of our study. A whole paragraph was added in page 3 and 4 of the manuscript: “ A total of 42 hospitalized patients were included in the study. Patients’ descriptive characteristics on admission are shown in Table 1, as entire cohort and stratified by lung injury status. Female patients represented the majority, of the study population (69.01%, n = 29). The mean age was 55.9 (SD: 20.3) years. Arterial hypertension and obesity were the most common comorbidities documented in the study (n = 21 and 14, respectively). Of note, in the studied population none of the patients had comorbidities such as rheumatoid arthritis, systemic lupus erythematosus, inflammatory bowel disease or other chronic immune-mediated/autoimmune diseases that may bias the interpretation of the lymphocyte subpopulations. Of note, none of these patients had a history of immuno-modulatory medication use, either.
Lung injury ≤ 50% was observed in 19 patients (55.48%) while 15 patients (44.12%) had greater injury (> 50%) of lung parenchyma. Mean (SD) IL-6 was 39.46 IU/ml (52.48) at admission. Mean (SD) C-reactive protein was 72.90 mg/L (70.98) and 27.93 mg/L (40.09) at day 1 and day 5 respectively. Mean (SD) values of PO2/FiO2 were 249.59 mmHg (142.52) at baseline and 209.29 mmHg (135.93) at day 5. Similar reductions were observed for the aforementioned variables between day 1 and 5 for both ≤ 50 and > 50% of lung injury.”
Comment #3
Materials and Methods: it is crucial for correct interpretation of data to disclose whether COVID-19 patients enrolled in the study had such comorbidities as rheumatoid arthritis, systemic lupus erythematosus, inflammatory bowel disease or other chronic immune-mediated/autoimmune diseases; these conditions may bias the correct interpretations of the lymphocyte subpopulations.
Answer to Comment #3
We would really like to thank the reviewer for the comment. In our cohort none of the patients had comorbidities such as rheumatoid arthritis, systemic lupus erythematosus, inflammatory bowel disease or other chronic immune-mediated/autoimmune diseases that could bias the interpretation of lymphocyte subpopulation measurements. Of note, none of these patients had a history of immunomodulatory medication use. Most common comorbidity in the entire cohort was arterial hypertension as shown in Table 1. In page 3 and 4, Results section was edited accordingly to include this comment (please see answer to comment 2) and add to a more complete presentation of our findings.
Comment #4
Maybe table 1 would be more interesting if data are split according to the extent of lung involvement (i.e., <50% vs >50%), adding a column with p values between the two patient groups.
Answer to Comment #4
We would really like to thank the reviewer for this comment. We have taken into consideration this comment. It should be noted though that with this analysis we would lose statistical significance power as the two generated groups (< and > 50% of lung involvement) are smaller than the already limited study group. Thus, we have categorized the patients in the descriptive Table 1 according to reviewer’s comment (page 4).
Comment #5
Likewise, it would be desirable to add in Table 1 the most common patient comorbidities and to check whether there are statistical differences between the two groups (<50 vs >50%). Previous studies have consistently shown that comorbidities have a negative impact on the outcome.
Answer to Comment #5
We found this comment an important add-on that could be included in our manuscript. As answered in comment #3, the most common comorbidity in our study was arterial hypertension. Similar to our reply to previous comments, the limited sample size of the study population would result in a statistical analysis without significance considering comorbidities and markers of disease severity and outcomes. We have added in Table 1 all comorbidities documented in our study population, as proposed by the reviewer. Of note, no significant association was found in the statistical analysis between comorbidities and ≤ 50 or > 50% lung injury. A comment was added in page 6: “Similarly, no significant association was found in the statistical analysis between comorbidities and ≤ 50 or > 50% lung injury (all p = NS, data not shown).”
Comment #6
Table 1, day 5. IL-6 has been measured in only two patients, this piece of information should be omitted because it does not have any significance.
Answer to Comment #6
This piece of information was omitted. Table 1 was updated accordingly.
Comment #7
Materials and Methods, page 3. How lung disease involvement on computed tomography (CT) was estimated should be detailed, e.g., was it calculated according to Chung et al (Radiology 2020;295:202–207) or else?
Answer to Comment #7
We would like to thank the reviewer for this important comment. Lung disease involvement on computed tomography (CT) was calculated by the physicians of the Radiology department of our hospital. “Calculation of lung involvement was made similarly to Chung et al. In this study each of the five lung lobes was assessed for the degree of involvement and classified as none (0%), minimal (1%–25%), mild (26%–50%), moderate (51%–75%), or severe (76%–100%). In our study, lung involvement was calculated based on 2 axes; 1) in the evaluation of the entire lung parenchyma, simple approximation is as follows: 25% of lung parenchyma referred to each one of the lower lobes while 15% referred to each upper and middle lobe, respectively. 2) classification of severity was defined as: minimum (<10%), moderate (11-25%), important (26-50%), severe (51-75%) and critical (>75%), respectively.” We have added this paragraph in the Materials and Methods section, in page 3 of the manuscript. In addition, the proposed citation was added in this paragraph; Chung et al (Radiology 2020;295:202–207).
Comment #8
Materials and Methods, page 3. What was the rationale for measuring lymphocyte subpopulations on day 5? The outcome of COVID-19 is highly variable, why did the Authors choose this time point for follow-up of lymphocyte subpopulations? The Authors should explain their motivation in Material and Methods.
Answer to Comment #8
We would like to thank the reviewer for this comment. From early days of the pandemic, it has been shown that worsening of COVID-19 occurs through day 7 to day 12 of the disease. In our cohort mean duration of symptoms (from disease onset) was 6.29 days. Most of the patients who deteriorated, did so by day 7. Therefore, rationale for measuring lymphocyte subpopulations on day 5 of hospitalization in our study, was to include the timeline of clinical deterioration and not to exceed the average length of stay, in order to reduce selection bias.
Comment #9
Do the Authors have data on lymphocyte subpopulations at patient discharge from the hospital?
Answer to Comment #9
Unfortunately, due to the limited resources and the lack of funding to this study we could not measure lymphocyte subpopulations at patient’s discharge.
Comment #10
Discussion: Although the Authors found an association between NK cells and lung involvement, the significance of this finding is not discussed.
Answer to Comment #10
We would like to thank the reviewer for the comment. We have added a paragraph in Discussion section, page 7, regarding this association found in our study. We added: “CD16+56 NK cell counts have been shown to decrease in COVID-19 disease (Deng, X., H. Terunuma, and M. Nieda, Exploring the Utility of NK Cells in COVID-19. Biomedicines, 2022. 10(5).). Cytolytic activity of NK cells and associated production of cytokines exert a key role in the defensive immunological response against SARS-CoV-2 infection (Jeyaraman, M., et al., Bracing NK cell based therapy to relegate pulmonary inflammation in COVID-19. Heliyon, 2021. 7(7): p. e07635.). This cytolytic activity of NK cells serves as a possible mechanism of progressive lung damage observed in the present study with the increase in NK count. Similarly, in a study (n = 32 patients) with severe SARS-CoV-2 infection, a raised proportion of mature NK cells and low T cell counts was observed (Varchetta, S., et al., Unique immunological profile in patients with COVID-19. Cell Mol Immunol, 2021. 18(3): p. 604-612.).”
However, it is mentioned in page 6, that after FDR correction, this association did not remain statistically significant.
Comment #11
Finally, the Authors report that none of the associations remained significant after FDR correction. How does this impact on the overall significance of the paper?
Answer to Comment #11
Due to FDR for correcting multiple testing error, all significant results were lost. As a result, we cannot declare that there is strong evidence for these associations, and we can characterize it as suggestive. The small number of patients included in this study resulted in a reduced power of our analysis. However, the significance observed in specific parameters prior to FDR correction shows a trend towards the association of specific lymphocyte subpopulations and markers of COVID-19 disease severity. We hypothesize that a greater number of patients will probably confirm our findings.
Comment #12
English revision is necessary.
Answer to Comment #12
We have performed an English language revision, where applicable.
Reviewer 3 Report
In this article, the authors report the results of a study investigating associations in lymphocyte subpopulations with disease severity markers in patients hospitalized with COVID-19. The authors report that CD16+CD556+ cells (NK cells) were associated with a higher risk of lung injury (>50% lung parenchyma) and that an increase in the difference in CD3CD4 and CD45RO cells between days 1 and 5 was associated with a decreased difference in CRP levels, presumably leading to recovery and immune homeostasis. Conversely, the CD45RARO difference was associated with an increase in the CRP difference between the same time points.
This is a well-written and interesting article trying to analyze associations between lymphocyte subpopulation changes and disease severity markers. Although the study is small (only 42 patients), the results are interesting and potentially helpful in understanding the relationships in alterations of lymphocyte subpopulations and disease severity and recovery. However, there are a few concerns/critiques that need to be addressed.
1. In the Materials and Methods section, the authors state that patients were “randomly” selected for inclusion in the study and that the data was part of a larger study. First, the manuscript does not indicate whether the patients were given an informed consent. This needs to be indicated. Second, was there any criteria in selecting these patients out of the larger study? If more patients were included, why not include a larger number?
2. What was the mortality in these patient cohort?
3. In Table 1, for day 5 it appears that there were only two patients with IL-6 results and 29 for CRP levels. What are the reasons for the difference from day 1? Had these patients been discharged from the hospital before day 5?
4. Was the distribution of the data analyzed? If so, please indicate.
5. Please describe the methodology used for measuring IL-6 and CRP.
6. Figure 1-4 do not have y-axis labels.
7. The second line of the last paragraph on page 2 says: “Adult patients with COVID-19 patients, hospitalized ….” Remove the second “ patients”, it should say “ Adult patients with COVID-19 hospitalized ….”
8. In the discussion, the authors state that a functional exhaustion of immune cells [40] leads to the induction of both cellular [41] and humoral immune responses [42]. Do they mean a reduction rather than induction?
Author Response
Reviewer 3
Comments and Suggestions for Authors
In this article, the authors report the results of a study investigating associations in lymphocyte subpopulations with disease severity markers in patients hospitalized with COVID-19. The authors report that CD16+CD556+ cells (NK cells) were associated with a higher risk of lung injury (>50% lung parenchyma) and that an increase in the difference in CD3CD4 and CD45RO cells between days 1 and 5 was associated with a decreased difference in CRP levels, presumably leading to recovery and immune homeostasis. Conversely, the CD45RARO difference was associated with an increase in the CRP difference between the same time points.
This is a well-written and interesting article trying to analyze associations between lymphocyte subpopulation changes and disease severity markers. Although the study is small (only 42 patients), the results are interesting and potentially helpful in understanding the relationships in alterations of lymphocyte subpopulations and disease severity and recovery. However, there are a few concerns/critiques that need to be addressed.
We thank the reviewer for the meaningful and helpful comments and recommendations.
Comment #1
In the Materials and Methods section, the authors state that patients were “randomly” selected for inclusion in the study and that the data was part of a larger study. First, the manuscript does not indicate whether the patients were given an informed consent. This needs to be indicated. Second, was there any criteria in selecting these patients out of the larger study? If more patients were included, why not include a larger number?
Answer to Comment #1
We would like to thank the reviewer for this comment. Each patient enrolled in this study provided a written informed consent. We added this information in page 3 of the manuscript under the section 2.1. Study population. There were no specific inclusion criteria in the study since all patients were COVID-19 positive and adults. We were unable to enroll a larger number of patients due to lack of resources and funding for flow cytometry assays. It has to be noted that both our cohort-study (University Hospital of Ioannina COVID Registry) and this sub-study have not received specific funding.
Comment #2
What was the mortality in these patient cohort?
Answer to Comment #2
Regarding the mortarilty rate in this cohort, we can report that there were 5 (five) deaths in a total of 42 patients during the hospitalization. Two out of the total deaths occurred during the first 5 days.
Comment #3
In Table 1, for day 5 it appears that there were only two patients with IL-6 results and 29 for CRP levels. What are the reasons for the difference from day 1? Had these patients been discharged from the hospital before day 5?
Answer to Comment #3
We would like to thank the reviewer for this important comment. IL-6 was not routinely calculated at day 5. This information has now been omitted from day 5 at Table 1. Fewer measurements of CRP levels are due to earlier discharge of these patients and the 2 deaths prior to day 5, as indicated in the answer of comment # 2.
Comment #4
Was the distribution of the data analyzed? If so, please indicate.
Answer to Comment #4
We thank the reviewer for this comment. Data distribution was performed, and suitable regression analyses were applied.
Comment #5
Please describe the methodology used for measuring IL-6 and CRP.
Answer to Comment #5
We would like to thank the reviewer for this comment. The Access IL-6 assay is a paramagnetic particle, chemiluminescent immunoassay executed on Beckman DXi 800 immunoassay analyzers. CRP is calculated by rate nephelometry (IMMAGE, Beckman Coulter). Comments were added to the manuscript in page 3, under the section of Materials and Methods.
Comment #6
Figure 1-4 do not have y-axis labels.
Answer to Comment #6
We thank the reviewer for pointing this out. We reshaped the violin plots according to reviewer’s comment. We assume that with new figures it is clearer to understand the fundamental differences between days 1 and 5. Former figures 1, 2 and 3 are now presented as Figure 1 in page 6, similarly former figure 4 is now presented as Figure 2, in page 7 . In new Figure 1, axis-y labels are numerical given in mg/L for CRP and % for lymphocyte subpopulations presented in the plots. Similarly, in new Figure 2 same labeling applies for CRP and Cell plots.
Comment #7
The second line of the last paragraph on page 2 says: “Adult patients with COVID-19 patients, hospitalized ….” Remove the second “ patients”, it should say “ Adult patients with COVID-19 hospitalized ….”
Answer to Comment #7
We would like to thank the reviewer for this remark. The 2nd “patients” was replaced with disease. In page 2, under the section 2.1. Study Population we added: ‘’Adult patients with COVID-19 disease, hospitalized between June and July 2021, were randomly selected for inclusion in the study.’’
Comment #8
In the discussion, the authors state that a functional exhaustion of immune cells [40] leads to the induction of both cellular [41] and humoral immune responses [42]. Do they mean a reduction rather than induction?
Answer to Comment #8
The Reviewer is correct. We would like to thank the reviewer for tracing this mistake. We have corrected into reduction in the revised manuscript, in page 8 under the section Discussion.
Reviewer 4 Report
The authors try to understand the biomarkers for COVID lung injury and progression of severity however they authors mentions that none of the results are significant due to various factors listed in the manuscript. There are no significant association made with the flow cytometry data and outcomes, and no significant observations concluded and hence can't be considered to be a manuscript which enhances the knowledge of the field.
Author Response
REVIEWER 4
Comment #1
The authors try to understand the biomarkers for COVID lung injury and progression of severity however they authors mentions that none of the results are significant due to various factors listed in the manuscript. There are no significant association made with the flow cytometry data and outcomes, and no significant observations concluded and hence can't be considered to be a manuscript which enhances the knowledge of the field.
Answer to Comment #1
We would like to thank the reviewer for his/her careful review of our manuscript. We appreciate your feedback and constructive criticism. We would like to clarify that our study was designed to explore potential biomarkers for COVID-19 lung injury and severity progression. We understand the importance of having significant observations and conclusions in scientific research and we appreciate your assessment of our work. However, we believe that our study still provides useful information in the field of cellular immunologic response to SARS-CoV-2 infection and contributes to the advancement of knowledge in an evolving area, based on the on the presented associations:
- CD16CD56 cells (Natural Killer cells) were associated with a higher risk of lung injury (>50% of lung parenchyma)
- Increase in CD3CD4 and CD4RO cell count difference between day-5 and day-1 resulted in a decrease of CRP difference between these timepoint and CD45RARO difference was associated with an increase in the difference of CRP levels between the two timepoints.
The lack of other significant differences in the rest of lymphocyte subpopulations in the presented data is discussed in the manuscript and may result from the small sample size or the specific characteristics of the analyzed population. The discussion has been edited, in order to include a comment on the complexity of immune responses during COVID-19 progression.
Round 2
Reviewer 1 Report
1) Include Mean(SD)/Frequency in the table 1
2) Did you mean naive (CD45RA) in line 263? If so, please correct it.
Author Response
REVIEWER 1
Comments and Suggestions for Authors
Comment #1
Include Mean(SD)/Frequency in the table 1
Answer to Comment #1
We would like to thank the reviewer for his/her comment. We have changed the headings in columns 2 and 3 of Table 1 to enhance clarity. N is the number of observations and the values in columns 3 to 5 represent either mean(SD) or frequency, which is specified in column one, next to each variable.
Comment #2
Did you mean naive (CD45RA) in line 263? If so, please correct it.
Answer to Comment #2
We would like to thank the reviewer for his/her comment. Indeed, we meant naïve CD45RA cells, and we have now corrected the sentence accordingly (line 268 of the manuscript).
Reviewer 2 Report
The Authors have amended their manuscript. However, a column with p values is still missing in Table 1 comparing data between <50% and >50% lung injury at day 1 and day 5. As an example, the levels of IL-6 at day 1 between <50% and >50% appear to differ significantly (27.43 vs 60.73, respectively), but there's no statistical calculation. The same holds true for P/F (342.67 vs 244.15). If the Authors cannot obtain statistical significance because of limited numbers of patients, this should be reported anyway, explained to the reader, and discussed as a study limitation. Likewise, the rationale for measuring lymphocyte subpopulations on day 5 should also be explained to the reader, not only to the reviewer. Therefore, the Authors should include this piece of information in the Methods section of the manuscript, as already suggested.
Author Response
REVIEWER 2
Comment #1
The Authors have amended their manuscript. However, a column with p values is still missing in Table 1 comparing data between <50% and >50% lung injury at day 1 and day 5. As an example, the levels of IL-6 at day 1 between <50% and >50% appear to differ significantly (27.43 vs 60.73, respectively), but there's no statistical calculation. The same holds true for P/F (342.67 vs 244.15). If the Authors cannot obtain statistical significance because of limited numbers of patients, this should be reported anyway, explained to the reader, and discussed as a study limitation.
Likewise, the rationale for measuring lymphocyte subpopulations on day 5 should also be explained to the reader, not only to the reviewer. Therefore, the Authors should include this piece of information in the Methods section of the manuscript, as already suggested.
Answer to Comment #1
We would really like to thank the reviewer for this comment. It is true that the subgroup analysis of patients based on the degree of lung injury did not yield statistically significant results (p was non-significant for all comparisons), owing to the small number of patients in each group. We have therefore clarified this by adding a comment in the results (line 183 to 185) and in the limitations (line 281 to 284): “Second, comparisons between subgroups of patients based on the degree of lung injury did not yield statistically significant results, owing to the small number of patients in each group. Third, for the same reasons, it was not possible to analyze outcomes such as death and intubation in this cohort of patients due to the small number of these events.”
Regarding the rationale for measuring lymphocyte subpopulations on day 5 we have added a comment in the Methods section (line 120 to 125 of the manuscript): ‘From the early days of the pandemic, it has been shown that worsening of COVID-19 occurs through day 7 to day 12 of the disease. Most of the patients who deteriorated, did so by day 7 [20]. In the study cohort mean duration of symptoms (from disease onset) was 6.29 days. Therefore, the rationale for measuring lymphocyte subpopulations on day 5 of hospitalization, as a second timepoint, was to include the timeline of clinical deterioration and not to exceed the average length of stay, to reduce selection bias.’’ In addition, the proper citation was added in this paragraph; Wang et al (JAMA. 2020 Mar 17; 323(11): 1061–1069).
Reviewer 4 Report
The authors have not made any significant changes to observations or conclusions. The study does not fit with the requirements for publication in this journal.
Author Response
REVIEWER 4
Comment #1
Comments and Suggestions for Authors
The authors have not made any significant changes to observations or conclusions. The study does not fit with the requirements for publication in this journal.
Answer to Comment #1
We appreciate the time and effort the Reviewer has devoted to our study. We have made every effort to ameliorate our manuscript, during the first and second revision, by adding important clarifications, presenting our results in a more detailed manner, addressing comments and concerns, and strengthening our discussion.